# Kynurenine Pathway after Kidney Transplantation: Friend or Foe?

**DOI:** 10.3390/ijms25189940

**Published:** 2024-09-14

**Authors:** Izabela Zakrocka, Ewa M. Urbańska, Wojciech Załuska, Andreas Kronbichler

**Affiliations:** 1Department of Nephrology, Medical University of Lublin, 20-093 Lublin, Poland; izabela.zakrocka@umlub.pl (I.Z.); wojciech.zaluska@umlub.pl (W.Z.); 2Department of Experimental and Clinical Pharmacology, Medical University of Lublin, 20-093 Lublin, Poland; ewa.urbanska@umlub.pl; 3Department of Internal Medicine IV, Nephrology and Hypertension, Medical University Innsbruck, 6020 Innsbruck, Austria; 4Department of Health, Medicine and Caring Sciences, Linköping University, 581 83 Linköping, Sweden

**Keywords:** kidney, transplantation, graft, rejection, immunosuppression, kynurenine, kynurenic acid, tryptophan, infection, cancer

## Abstract

Kidney transplantation significantly improves the survival of patients with end-stage kidney disease (ESKD) compared to other forms of kidney replacement therapy. However, kidney transplant recipients’ outcomes are not fully satisfactory due to increased risk of cardiovascular diseases, infections, and malignancies. Immune-related complications remain the biggest challenge in the management of kidney graft recipients. Despite the broad spectrum of immunosuppressive agents available and more detailed methods used to monitor their effectiveness, chronic allograft nephropathy remains the most common cause of kidney graft rejection. The kynurenine (KYN) pathway is the main route of tryptophan (Trp) degradation, resulting in the production of a plethora of substances with ambiguous properties. Conversion of Trp to KYN by the enzyme indoleamine 2,3-dioxygenase (IDO) is the rate-limiting step determining the formation of the next agents from the KYN pathway. IDO activity, as well as the production of subsequent metabolites of the pathway, is highly dependent on the balance between pro- and anti-inflammatory conditions. Moreover, KYN pathway products themselves possess immunomodulating properties, e.g., modify the activity of IDO and control other immune-related processes. KYN metabolites were widely studied in neurological disorders but recently gained the attention of researchers in the context of immune-mediated diseases. Evidence that this route of Trp degradation may represent a peripheral tolerogenic pathway with significant implications for transplantation further fueled this interest. Our review aimed to present recent knowledge about the role of the KYN pathway in the pathogenesis, diagnosis, monitoring, and treatment of kidney transplant recipients’ complications.

## 1. Introduction

Kidney diseases represent a broad spectrum of disorders with various etiologies and clinical presentations. Although the number of individuals worldwide affected by kidney diseases is estimated at 850 million, their true prevalence seems to be largely underestimated [1]. It was suggested that kidney disorders are one of the most common healthcare problems, as the number of people affected by kidney diseases is significantly larger than patients with diabetes or individuals infected by the human immunodeficiency virus (HIV) [1]. Chronic kidney disease (CKD), the predominant kidney disorder with a significant impact on health outcomes, often diagnosed late or unrecognized, is expected to be the 5th leading cause of death by 2040 [2]. Patients with CKD are at exceptionally high risk of cardiovascular events, largely dependent on preexisting comorbidities and the presence of specific risk factors [3,4]. The incidence of cardiovascular events was reported to be even higher in advanced CKD, namely end-stage kidney disease (ESKD), resulting in a 5-year survival rate in patients receiving kidney replacement therapy (KRT) of only 48% in Europe and 41% in the United States of America [5]. According to the European Renal Association Registry, the unadjusted incidence of KRT in Europe in 2021 was 145 per million, whereas the number of kidney transplantations performed per million was estimated at 37 [6]. Among KRT methods, kidney transplantation represents the gold standard of treatment for ESKD, resulting in a substantial improvement in patients’ 5-year survival, ranging from 85.8% (deceased donor) to 94.2% (live donor) [7]. The unadjusted 5-year graft survival was shown to be lower, 77.0% in deceased donor transplantations vs. 88.3% in live donor kidney transplantations [7]. In Europe, despite a lower transplant rate than in America, more ESKD patients live with a functioning graft (47% vs. 32%, respectively). This can still be considered unsatisfactory [8]. Despite the improvements in surgical techniques, organ preservation methods, and immunosuppressive modalities, patient and graft survival are limited, mainly due to immunological damage of the transplanted organs, as well as cardiovascular, infectious, and oncological complications [9,10]. Searching for reliable and well-validated biomarkers is of special importance in kidney transplant recipients’ care [11]. Proposed biomarkers, including plasma donor-derived cell-free DNA (dd-cfDNA), blood Torque teno virus copy numbers, or urinary chemokine tests, are promising noninvasive methods of graft function evaluation already under investigation [12,13]. However, the increasing number of kidney transplant recipients necessitates the search for novel strategies for the diagnosis and treatment of transplant-related complications.

Tryptophan (Trp) is one of the eight essential amino acids. Yet, only 1% of its pool is used for protein synthesis [14]. The remaining 99% of available Trp is metabolized into biologically active substances, including serotonin (1–2%), and several molecules formed along the kynurenine (KYN) pathway [15]. The KYN pathway, a major route of Trp degradation, has been extensively studied in the context of neurological [16,17], inflammatory [18], oncological [19], kidney [20,21], cardiovascular [22,23], and metabolic disorders [24,25]. The functional link between Trp metabolism and organ transplant complications received increasing attention during the last few years. The Trp-KYN pathway was implicated in the host defense against pathogens and in peripheral mechanisms of tolerance induction, mainly through its impact on T cell activity [26]. 

In this review, we discuss the putative role of the KYN pathway in immunological control after kidney transplantation and in the risk of infectious, oncological, and cardiovascular complications. Additionally, diagnostic and therapeutic implications for kidney transplant recipients’ management are highlighted. 

## 2. The Kynurenine (KYN) Pathway

Trp degradation begins with a rate-limiting step, which is catalyzed by 3 enzymes: indoleamine-2,3-dioxygenase 1 (IDO1), indoleamine-2,3-dioxygenase 2 (IDO2), or tryptophan-2,3-dioxygenase (TDO) (Figure 1). 

All Trp-degrading enzymes are hemoproteins [27]. TDO is exclusively expressed in the liver and central nervous system and controls the entrance of Trp into the KYN pathway under physiological conditions [28]. Due to its low affinity for Trp, TDO is active when Trp concentrations exceed those required for protein or serotonin synthesis, indicating the role of TDO in the prevention of Trp accumulation [15]. Several factors, including Trp [29], glucocorticoids (GCs), and glucagon [30,31], were shown to upregulate TDO activity. In contrast to TDO, IDO is an inducible enzyme widely distributed in various cells; in immune cells, it includes monocytes, macrophages [32], and dendritic cells (DCs) [33]. IDO gene transcription is mainly stimulated by interferon-γ (IFN-γ) [34], tumor necrosis factor-α (TNF-α), interleukin-1 (IL)-1 [35], and toll-like receptors (TLRs) activated by lipopolysaccharide (LPS) [36]. However, IDO expression should not be considered equal to IDO’s activity, since some DCs may constitutively express IDO and require stimulatory signals (i.e., superoxide) for posttranslational heme modification to activate IDO [37]. During the activation of IDO by immunological stimuli, the activity of TDO is substantially suppressed, and IDO remains the main enzyme responsible for Trp breakdown [38]. Additionally, IDO has a lower capacity with a significantly higher affinity for Trp compared to TDO [39]. Two distinct isoforms of IDO have been found in humans. Whereas IDO1 seems to play a major role in Trp depletion, resulting in antimicrobial defense and T cell suppression, and most available data regarding IDO are related to IDO1, the role of IDO2 is much less explored [40]. Several studies suggested IDO2 involvement in oncogenesis [41] through the inhibition of immune response to tumor cells [42]. However, other Trp-degrading enzymes, TDO and IDO1, have also been found on cancer cells, pointing to their role in facilitating the immune escape of tumors [28]. 

After the transformation of Trp to N-formylkynurenine (N-formyl KYN) by IDO or TDO, the next step entails the formation of KYN, which later can be converted into three types of metabolites. KYN is preferentially metabolized into 3-hydroxykynurenine (3-HK) by kynurenine-3-monooxygenase (KMO) and then by kynureninase (KYNU) into 3-hydroxyanthranilic acid (3-HAA), which is a substrate for picolinic acid (PA) or quinolinic acid (QA) production, with nicotinamide adenine dinucleotide (NAD^+^) as a final product [43]. Although some data show that most Trp metabolites from this branch, especially 3-HK, 3-HAA, and QA, act as antioxidants [44], the majority of reports indicate that they exert pro-oxidant activity [45,46,47] and thus may evoke cell death [48]. IDO/TDO and KMO enzymes are crucial to the control of Trp degradation and its metabolic effects.

Kynurenic acid (KYNA), synthesized from KYN by kynurenine aminotransferases (KATs), is a broad-spectrum antagonist of all types of ionotropic glutamate (GLU) receptors of N-methyl-D-aspartate (NMDA) [49], alpha-amino-3-hydroxy-5-methyl-4-isoxazolepropionic acid (AMPA), and kainate [50] type. Neuroprotective, antiepileptic, and neuromodulatory effects of KYNA are well described, and the possible impact of KYNA on brain disorders is broadly accepted [51,52,53,54]. KYNA was also shown to inhibit α7 nicotinic acetylcholine receptors [55] and to act as a ligand for aryl hydrocarbon receptor (AhR) [56] and G protein-coupled receptor 35 (GPR35), which regulate the immune response [57]. Consequently, through the AhR- [58] and GPR35-mediated processes, KYNA may act as an immunosuppressive agent [59] and a guardian of gut microbiota homeostasis, linking the immune system with microbiota and metabolic disorders [60,61]. 

The third branch of the KYN pathway, yielding anthranilic acid (AA), is the least studied route of Trp metabolism. Since AA can be converted through nonspecific hydroxylation into 3-HAA (Figure 1), it becomes a precursor for neuroactive products. Although AA itself presented ambiguous properties in the context of neurological disorders, a complex interplay between AA and the gut-brain axis was recently highlighted [62].

Trp and its metabolites have been extensively studied in CKD patients. KYN pathway activation, presented as IDO induction, and high serum KYN, KYNA, and QA levels correlate with CKD severity and inflammatory parameters, high-sensitivity C-reactive protein (hsCRP), and soluble TNF-receptor-1 (sTNFR-1) concentration, independently of serum creatinine level [63]. Additionally, KYN and its metabolites have been linked to oxidative stress [64], endothelial dysfunction [65], cognitive impairment [66], and incident cardiovascular disease in CKD patients [67,68]. Although products of Trp degradation can accumulate in the body secondary to impaired glomerular filtration, their level, especially in the serum, was shown to tightly correlate with immune system activation and possibly to predict kidney damage and its progression. The measurement of serum IDO activity together with the Naples prognostic score has been recently proposed as a useful tool in early CKD diagnosis [69]. 

Importantly, Trp is present at a 2 mmol/L concentration in one of the most popular preservation solutions, a histidine-tryptophan-ketoglutarate (HTK) solution, mainly because of its antioxidant and membrane-stabilizing potential [14]. HTK, invented by Bretschneider, was shown to be a comparable preservation solution when preservation times were shorter and when standard criteria donor organs were implanted [70,71]. Modified HTK solution, a HTK-N solution, was shown to lower endothelial graft damage, although observations in kidney graft recipients are limited [72]. On the other hand, Trp deprivation is suggested to induce resistance to hypoxia and prevent ischemia-reperfusion injury, mainly by lowering the number of circulating neutrophils [73] or autophagy induction, which may decrease the secretion of proinflammatory cytokines [74]. Indeed, Eleftheriadis et al. reported that Trp deprivation increased renal proximal tubular epithelial cell survival in hypoxic conditions through autophagy induction [75]. Further studies are needed to clarify the impact of Trp on kidney graft preservation results. 

## 3. Immune Function in Kidney Transplantation and the Kynurenine Pathway

Despite significant progress in immunosuppressive treatment and methods of immunological monitoring, the 10-year graft survival rate remains suboptimal, reaching 53.6% in patients with kidney transplants received from deceased donors [76]. Chronic antibody-mediated rejection (ABMR) is considered the most common cause of graft loss, without clear benefits from currently available treatment strategies, including intensified immunosuppression or therapeutic plasma exchange (TPE). Therefore, it is of special importance to search for novel targets for the monitoring and treatment of immune-mediated kidney graft damage. 

### 3.1. Indoleamine 2,3-Dioxygenase (IDO)

IDO remains one of the most extensively studied enzymes in the KYN pathway. The interest in this field has rapidly increased since Munn et al. showed that female mice accepted semi-allogenic fetuses under undisturbed Trp catabolism only [77]. Due to pleiotropic immunoregulatory effects revealed through the years, IDO is considered one of the main guardians of immune tolerance, contributing to the downregulation of allogenic responses (Figure 2). 

IDO is expressed constitutively in the majority of cells and tissues, including immunocompetent cells such as monocytes, macrophages, and DCs. In the kidney, IDO expression was mainly detected in renal tubular epithelial cells [78], which are critically involved in kidney graft rejection, as well as in mesangial [79] or glomerular endothelial cells, as shown in patients with graft glomerulitis [80]. However, IDO localization in the graft may vary through time. Considerable IDO expression was detected around atrophic tubules, in cells resembling endothelial and mesangial cells, and in minor quantities in tubular epithelial cells in biopsy samples obtained from patients at two years after kidney transplantation and with signs of chronic damage [81]. IDO activity is highly susceptible to stimulation with pro-inflammatory stimuli, such as IFN-γ, TNF-α, and TLRs, whereas anti-inflammatory cytokines IL-4 [82], IL-10, transforming growth factor-β (TGF-β) [83], and nitric oxide (NO) [84] diminish it. Among the possible effects of IDO on the immune system, its influence on T cells was widely explored. Stimulation of IDO resulting in Trp depletion was shown to activate the general control non-derepressible 2 (GCN2) kinase pathway, causing an arrest of T cell proliferation in the mid-G1 phase and making them susceptible to apoptosis [85]. Interestingly, apoptosis of T helper (Th1) cells rather than Th2 cells may be evoked by KYN metabolites, 3-HAA and QA [86]. As shown in mesenchymal stem cells, IFN-γ-induced stimulation of IDO results in regulatory T cell (Treg) induction, contributing to local immunosuppressive effects and to attenuation of renal fibrosis [87]. Ligation of B7-1/B7-2 molecules by cytotoxic T-lymphocyte-associated protein 4 (CTLA4)/CD28 present on the surface of Tregs or by a separate CTLA4 fusion protein (CTLA4-Ig) is necessary to activate IDO and afterward inhibit T cell proliferation [88].

IDO activation was also suggested to increase macrophage polarization into the M2 phenotype [89], suppress natural killer (NK) cell activity [90], decrease mean histocompatibility complex (MHC) class I gene expression [91], and be implicated in the development of local immune tolerance. Although IDO does not affect B cell proliferation [92], there is some evidence suggesting the impact of IDO on B cell-driven responses [93]. Inhibition of the mammalian target of rapamycin (mTOR) is another potential mechanism related to IDO’s action [94].

Furthermore, IDO1 may modulate immunological system activity through other mechanisms. The activation of AhR, a transcription factor crucial for immune system control, increases both IDO1 and IDO2 expression in DCs, together with the generation of FoxP3(+) Tregs [95]. On the other hand, it was demonstrated that IDO products alone, especially KYN, can activate AhR and lead to Treg generation [96]. Moreover, in plasmacytoid DCs, TGF-β-dependent self-tolerance was linked with the non-enzymatic regulatory activity of IDO [97]. 

There is ongoing debate about the significance of the KYN pathway activity measurement in kidney transplant recipients. The key question remains whether elevated Trp degradation and production of KYN-derived compounds is an immunosuppressive mechanism preventing tissue damage or rather reflecting already ongoing tissue breakdown. Discrepancies come from different models tested (animal vs. human), diverse tissues and body fluids analyzed (serum, urine, or graft biopsy samples), or analytical methods. Another issue that should be taken into consideration is the kidney transplant recipient’s condition. Lower serum Trp, which is considered a marker of IDO activation, may not reflect true amino acid turnover. After kidney transplantation, based on improved protein ingestion and kidney function normalization, an elevation of serum Trp level should be expected [98]. Indeed, it has been shown that already during the first 3 days after transplantation, changes in Trp and its metabolites can be observed, with a significant reduction of serum KYNA and KYN and an increase of Trp [99]. On the other hand, higher serum Trp can be expected in the case of tissue damage, especially under inflammatory stimuli [100]. The role of gut microbiota in protein breakdown and KYN metabolite formation should not be forgotten. Therefore, the diagnostic and prognostic relevance of the KYN pathway monitoring in kidney transplant recipients remains a huge challenge.

In one of the first studies in kidney transplant recipients, Holmes et al. reported that the serum KYN level started to increase 5 to 7 days prior to a biopsy-confirmed acute rejection (AR) episode, and it did not correlate with serum creatinine concentration or high dose steroid treatment [101]. Later on, it was shown that serum levels of KYN and Trp are higher in non-rejecting allograft recipients than in healthy volunteers, and that their levels considerably increase during rejection episodes [102]. Importantly, the serum KYN/Trp ratio in the rejection group was higher starting from day 1 after transplantation and correlated with serum neopterin concentration, a marker of immune system activation [103]. Moreover, kidney biopsy specimen staining indicated IDO upregulation in tubular epithelial cells of rejected grafts, whereas in healthy grafts, IDO expression was not detectable. Similarly, it was reported that the serum KYN level increases from the 3rd day before biopsy-proven kidney graft rejection, as compared to recipients without rejection episodes [104]. It remains unclear whether the KYN pathway activation can be related to a specific type of graft rejection. In a recently published study, a higher IDO1 expression in interstitial foci of kidney graft biopsies specimens was related to a higher incidence of ABMR (15% vs. 2.44%, *p* = 0.035) and T cell-mediated rejection (TCR) (53.75% vs. 17.07%, *p* = 0.0001) [105]. The significance of IDO elevation in kidney transplant recipients exposed to a plethora of immune stimuli is not fully clarified [106]. Studies in pediatric kidney graft recipients suggested that serum KYN/Trp ratio elevation can be an important tool in differentiating acute graft rejection from infection [107,108]. 

### 3.2. Kynurenine 3-Monooxygenase (KMO)

Possible involvement of KMO in graft survival was recently highlighted. In contrast to IDO, KMO is a mitochondrial outer membrane protein, found preferentially in tubular epithelial cells [109], glomerular cells, and podocytes [110]. KMO is an important enzyme of the KYN pathway, which controls the amount of KYN entering three different arms of the path and thus regulates the formation of various biologically active products, including NAD^+^ (Figure 1). Diminished de novo NAD^+^ formation due to KMO downregulation in proximal tubular epithelial cells was postulated as one of the potential mechanisms of acute kidney injury (AKI) [111]. Direct KMO products, 3-HK together with 3-HAA, are well-known immunosuppressive agents. Both compounds significantly inhibit T cell proliferation and survival without effect on DCs or endothelial cells, as shown in the murine corneal allograft model [112]. Inhibition of KMO, resulting in lower 3-HK and 3-HAA formation, can be related to a higher immunological risk of graft rejection. Indeed, in a porcine model of acute kidney allograft rejection, lower KMO gene transcription and protein level, with concomitant higher IDO gene expression and enzyme activity, were demonstrated [113]. Similarly, a significant reduction in KMO activity was observed in a pig model of kidney allograft rejection [109]. Additionally, 3-HK and 3-HAA protected tubular epithelial cells from injury, in part through inhibition of T cell proliferation and upregulation of AhR expression. Moreover, 3-HK was reported to inhibit mesangial cell proliferation [114], whereas 3-HAA and 3-HK attenuated glomerular injury in an experimental autoimmune glomerulonephritis animal model [115]. 

On the other hand, in animals with the KMO knockout, a proteinuria phenotype developed, and in diabetic patients, a significant decrease in podocytic KMO expression was found [110]. Thus, more studies are needed to explore the role of KMO and its metabolites in kidney diseases.

### 3.3. Kynurenic Acid (KYNA)

The immunomodulatory role of KYNA is well recognized. Through the impact on GPR35, KYNA may inhibit the activation of the NLRP3 inflammasome in macrophages and suppress the production of caspase-1 and IL-1 β [116]. KYNA, through an activation of AhR, was shown to enhance the expression of TNF-stimulated gene 6 (TSG-6) in human mesenchymal stem cells and to reduce immune cell infiltration in an animal model of acute lung injury [58]. The potential impact of KYNA on T cell activity should also be considered, since AhR activation promotes differentiation of Tregs and inhibits expression of proinflammatory IL-17 [117]. Indeed, KYNA was reported to decrease IL-23 secretion by LPS-activated DCs and Th17 cell polarization [118]. Additionally, KYNA lowered human invariant NK T cell activation through GPR35 [119]. However, in an animal model of arthritis, KYNA failed to promote Treg development, contrary to the other NMDA antagonist memantine [120]. Other effects, including inhibition of TNF-α secretion from human mononuclear cells and high mobility group box protein 1 (HMGB1) secretion in monocytes and granulocyte cultures, were shown [121]. Although KYNA was not examined in kidney transplant patients in the context of immunological risk, presented data suggest the involvement of KYNA in immune tolerance formation. Interestingly, it was reported that tacrolimus (TAC), but not cyclosporine (CsA), elevates KYNA formation, suggesting KYNA’s role in selected immunosuppressive drug action [122]. 

## 4. Infections 

In addition to the great impact on alloantigen tolerance induction, the activation of the KYN pathway may impair the immune response against pathogens, which is of special importance in kidney transplant recipients. Trp degradation is a well-described mechanism of suppressing pathogens’ growth, including *Chlamydia* species [123] or cytomegalovirus (CMV) [124]. However, an uncontrolled KYN pathway activation, especially during sepsis, is considered to act bidirectionally. The pro-inflammatory state results in an increased level of certain Trp metabolites and leads to an impaired response to opportunistic infections [125], which may have a tremendous impact on patient outcomes. Indeed, plasma KYNA levels in patients with septic shock correlated with lactate and procalcitonin concentration only in survivors. Furthermore, a lack of reduction of KYNA after continuous veno-venous hemofiltration (CVVH) procedures was suggested as a factor predicting the fatal outcome of septic shock [126]. Although data from kidney transplant recipients are limited, KYN pathway activity analysis was indicated as a promising tool in early diagnosis of infectious complications and evaluation of their intensity. In a group of 355 kidney graft recipients, blood KYN levels were positively correlated with CMV infection severity prior to the diagnosis and were higher in patients with bacterial sepsis [106]. A decrease in blood KYN concentration reflected a response to the treatment, whereas its further increase was associated with poor outcomes. Similarly, serum KYN levels were markedly elevated in kidney transplant recipients with viral and Gram-negative bacterial infections in the absence of AR, making this parameter useful in differentiating posttransplant complications [101]. Interestingly, KYN and QA plasma concentrations were shown to increase significantly during CMV, but not BKV infection, indicating their significance in the fast detection of systemic immune activation linked with CMV [127]. More studies are needed to explore the relevance of KYN pathway metabolite measurements in different types of infections after kidney transplantation and its impact on long-term outcomes. 

## 5. Malignancies

Despite the well-known effects of KYN pathway activation in immune tolerance, Trp degradation can be a double-edged sword regarding carcinogenesis, especially in kidney transplant recipients. In the tumor microenvironment, a complex interplay between immune and cancer cells regulates tumor growth, its response to treatment, and metastasis formation. In clear cell renal cell carcinoma (RCC), secreted IL-10 and TGF-β promote Treg differentiation, which suppresses effector T cells and stimulates tumor growth [128]. Increased PD-L1 (programmed death-ligand 1) expression is another mechanism of negative T cell function control by cancer cells [129]. It was demonstrated that PD-L1 expression is enhanced by IDO1 in cancer cells in an AhR-dependent manner [130] and reduced by IDO1/TDO inhibition [131]. Co-expression of PD-L1 and IDO1 are negative prognostic factors associated with poor overall survival [132]. IDO1 and IDO2 were found in various types of cancers [133], and their presence was related to resistance to therapy and lower survival. A higher KYN/Trp ratio in clear cell RCC patients was associated with a decreased 5-year survival compared to subjects with low KYN/Trp values [134]. Other abnormalities in KYN pathway activity in tissues of RCC patients include lower expression of QPRT and downregulation of KMO and 3-HAAO [135], although QPRT activity may vary in different types of cancers [136]. Under QPRT inhibition, NAD^+^ synthesis is secured by the activation of a salvage pathway, often preferred by cancer cells, in which nicotinamide is converted by nicotinamide phosphoribosyltransferase (NAMPT) into NAD^+^ [137]. In patients with metastatic RCC, increased tissue TDO expression was strongly associated with higher KYN levels, a worse response to immunotherapy, and higher mortality [138]. Therefore, the KYN pathway can be an interesting target for diagnosis, monitoring, and treatment of malignancies [139]. In fact, IDO [140], TDO [141], or KMO [142] inhibitors were found effective as antineoplastic agents in certain types of cancer [143]. However, inhibitors of the KYN pathway enzymes should be used with great caution in kidney transplant recipients due to the potential risk of AR, as observed with programmed cell death protein 1 (PD-1), PDL-1, and CTLA-4 inhibitors [144]. On the other hand, KYNA exhibited antiproliferative effects on glioblastoma [145], colon [146], and renal [147] cancer cells, but its potential role in carcinogenesis is not fully clarified [148]. Therefore, a more selective KYN pathway inhibition should be preferred. 

## 6. Cardiovascular Risk

KYN pathway metabolites play a complex role in cardiovascular disease pathogenesis. Although Trp degradation products possess anti- and pro-inflammatory properties, most studies correlated them with higher cardiovascular risk, especially due to AhR activation [23]. Recent animal studies suggested a direct effect of KYN on blood pressure elevation and kidney damage [149]. Elevated Trp breakdown was associated with the development of cardiovascular disease [22] and was suggested as a prognostic factor in heart failure [150], coronary artery disease [151], pulmonary hypertension [152], and thromboembolic episodes [153], especially in patients with established CKD. In a recent Chronic Kidney Disease–Renal Epidemiology and Information Network (CKD-REIN) study, KYN was established as an independent risk factor for nonfatal and fatal cardiovascular outcomes in patients with CKD [154]. In this cohort of patients, increased Trp catabolism was correlated with procoagulant factors including plasma tissue factor [155], prothrombin fragments 1 + 2 [156], thrombomodulin, and von Willebrand factor level [157]. Although more evidence indicates the involvement of KYN metabolites in tissue damage, KYNA displays distinct properties. In bovine endothelial cell cultures, KYNA prevented cells from homocysteine-induced impairment, significantly increasing their migration and proliferation [158]. A positive correlation between KYNA plasma concentration and hyperhomocysteinemia in peritoneal dialyzed patients with atherosclerotic cardiovascular disease, possibly reflecting the protective effects of KYNA in this group of patients, was reported [159]. A relationship between serum KYNA level and aortic stiffness in patients with atrial fibrillation was also presented [160].

In kidney transplant recipients, cardiovascular diseases remain the most common cause of death, accounting for about 20–35% of overall mortality and significantly decreasing life expectancy [161]. Therefore, the search for novel diagnostic and therapeutic targets in this population is of special importance. Due to the fact that serum KYN and 3-HK levels were shown to be independently associated with graft failure, and 3-HK level was linked with higher mortality in kidney transplant recipients [100], more studies are necessary to explore the impact of the KYN pathway activation on cardiovascular risk and mortality in this particular group of patients. 

A summary of possible effects of KYN pathway activity alterations in kidney transplant recipients is presented in Table 1. 

## 7. Pharmacological Interventions

The impact of pharmacotherapy, in particular immunosuppression, on the KYN pathway activity after kidney transplantation is a matter of debate (Figure 3).

Since increased Trp metabolism along the KYN pathway induces immune tolerance, the impact of immunosuppressive drugs on the pathway seems conceivable. On the other hand, immunosuppressive agents can potentially restrict IDO overactivation. Changes in the KYN pathway activity may also serve as a diagnostic tool assessing the response to the pharmacotherapy, similarly to selected soluble checkpoint inhibitors in patients with antineutrophil cytoplasmic antibody (ANCA)-associated vasculitis (AAV) [162]. 

So far, the data on immunomodulating agents and the KYN pathway in kidney transplantation are rather limited. In phytohemagglutinin-stimulated and unstimulated human peripheral blood mononuclear cells, TAC, CsA, mammalian target of rapamycin inhibitor (mTORi) sirolimus, and methylprednisolone dose-dependently inhibited Trp degradation and neopterin level, whereas mycophenolate mofetil (MMF) was efficacious only at higher concentrations [163]. Different results were obtained in a mesangial cell line, in which incubation with MMF, CsA, and TAC significantly enhanced IDO expression, but sirolimus showed no effect [79]. As previously mentioned, TAC was shown to increase KYNA synthesis in cortical slices, which suggests that this KYN metabolite may be involved in TAC’s action [122]. Prior studies suggested that immunosuppressive drugs may affect the KYN pathway differently in various types of cells and that the outcome depends on coexisting immune stimuli. Studies analyzing the impact of CTLA4-Ig, abatacept and belatacept, on the KYN pathway did not yield consistent results. Abatacept failed to change the expression of IDO mRNA and KYN levels in DCs [164], and CTLA4-Ig did not increase the KYN/Trp ratio in bone marrow recipients [165]. In contrast, recombinant soluble CTLA4 induced the expression of IDO in human peripheral blood mononuclear cells and DCs [166]. Transduction of adenovirus-mediated CTLA4Ig (AdCTLA4Ig) increased IDO mRNA and KYN levels, and both effects were blocked by the IDO inhibitor 1-methylTrp (1-MT) [167]. In kidney graft biopsies specimens, belatacept treatment increased the number of Tregs in dysfunctional kidneys compared to CsA, whereas no changes in peripheral Tregs were observed [168]. Similarly, higher Treg levels, lower CD4(+)/IL-17A cell levels, decreased kidney graft fibrosis [169], and higher peripheral Bregs, Tregs, and DCs counts [170] were demonstrated after belatacept treatment in comparison to patients receiving CsA. Although an important alternative to calcineurin inhibitors, beletacept use is still limited due to the increased risk of AR and posttransplant lymphoproliferative disorder [171].

On the other hand, targeted KYN pathway inhibition may provide beneficial effects on the treatment of selected complications after kidney transplantation, namely malignancies or cardiovascular diseases. Beyond KMO, KATs, and KYNU inhibitors, which are potentially novel candidates in anticancer therapy [136], other already available drugs were shown to selectively inhibit the KYN pathway. Angiotensin converting enzyme inhibitors [172], angiotensin II type 1 receptor blockers [173], cyclooxygenase inhibitors [174], fibrates [175], and glibenclamide [176] were reported to inhibit KATs activity, which may result in the reduction of kidney damage-related complications, especially in kidney transplant recipients.

## 8. Conclusions

Generalized KYN pathway activation with the goal to evoke peripheral immune tolerance poses a risk of uncontrolled immune system depression; therefore, methods of its localized activation are of special interest. Local administration of IDO by viral vectors [177,178] or modulation of urinary microbiota [179] are interesting options for future studies on the modulation of KYN pathway activity in kidney transplant recipients. 

Exploring KYN pathway activity in patients after kidney transplantation offers an interesting approach to the diagnosis and treatment of tissue graft rejection and related complications. Due to the multiple and often opposing effects of the KYN pathway metabolites on cellular survival and immune processes, there is a need for enzyme- and target-organ-selective pharmacological tools. Precise targeting of the pathway may bring novel diagnostic and therapeutic tools, allowing better management of kidney transplant recipients. 

## Figures and Tables

**Figure 1 ijms-25-09940-f001:**
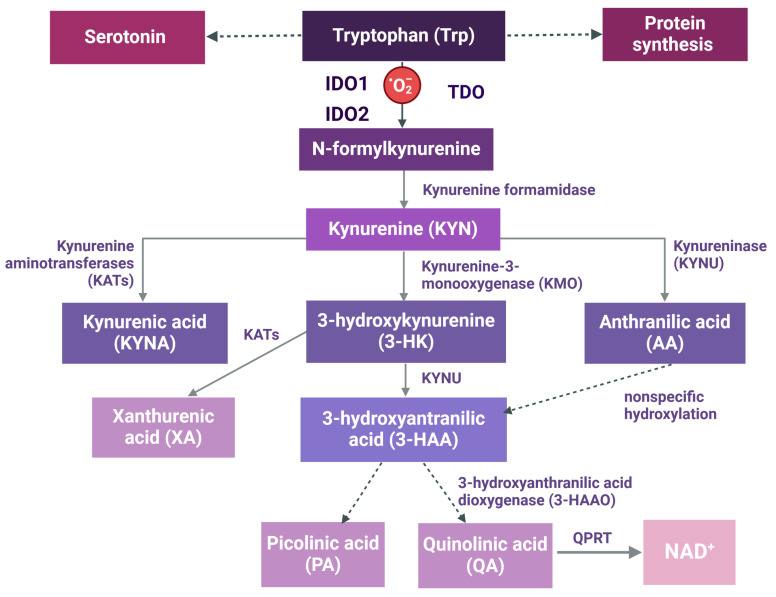
The kynurenine pathway. 3-HAA, 3-hydroxyanthranilic acid; 3-HK, 3-hydroxykynurenine; AA, anthranilic acid; 3-HAAO, 3-hydroxyanthranilic acid dioxygenase; IDO1, indoleamine-2,3-dioxygenase 1; IDO2, indoleamine-2,3-dioxygenase 2; KATs, kynurenine aminotransferase; KMO, kynurenine-3-monooxygenase; KYN, kynurenine; KYNA, kynurenic acid; KYNU, kynureninase; NAD^+^, nicotinamide adenine dinucleotide; PA, picolinic acid; QA, quinolinic acid; QPRT, quinolinic acid phosphoribosyltransferase; TDO, tryptophan-2,3-dioxygenase; Trp, tryptophan; XA, xanthurenic acid. Created with https://www.BioRender.com (accessed on 1 August 2024).

**Figure 2 ijms-25-09940-f002:**
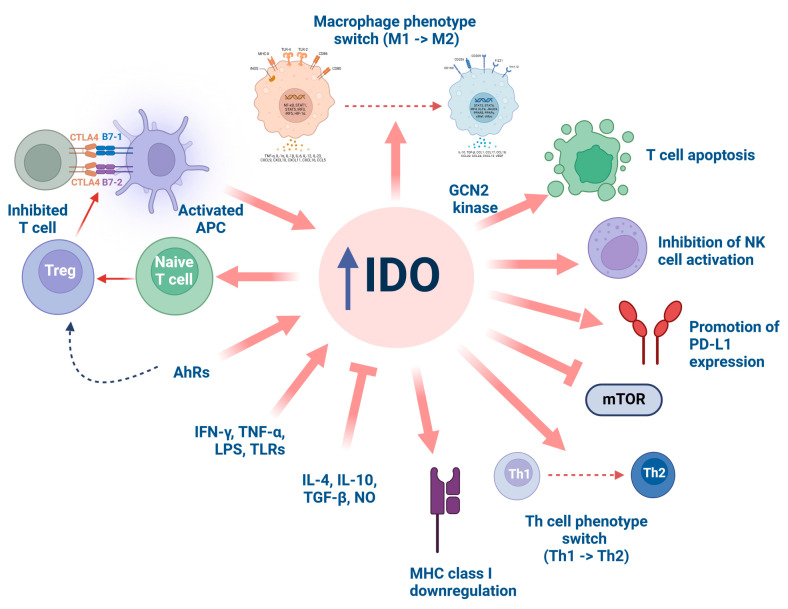
IDO activation resulting in immune cell activity modifications. AhR, aryl hydrocarbon receptor; APC, antigen-presenting cell; CTLA4, cytotoxic T-lymphocyte associated protein 4; GCN2, general control non-derepressible 2; IDO, indoleamine 2,3-dioxygenase; IFN-γ, interferon-γ; IL-4, interleukin-4; IL-10, interleukin-10; LPS, lipopolysaccharide; MHC, major histocompatibility complex; mTOR, mammalian target of rapamycin; NK, natural killer; NO, nitric oxide; PD-L1, programmed death-ligand 1; TGF-β, transforming growth factor-β; Th, T helper cell; TLR, toll-like receptor; TNF-α, tumor necrosis factor-α; Treg, regulatory T cell. Created with https://www.BioRender.com (accessed on 1 August 2024).

**Figure 3 ijms-25-09940-f003:**
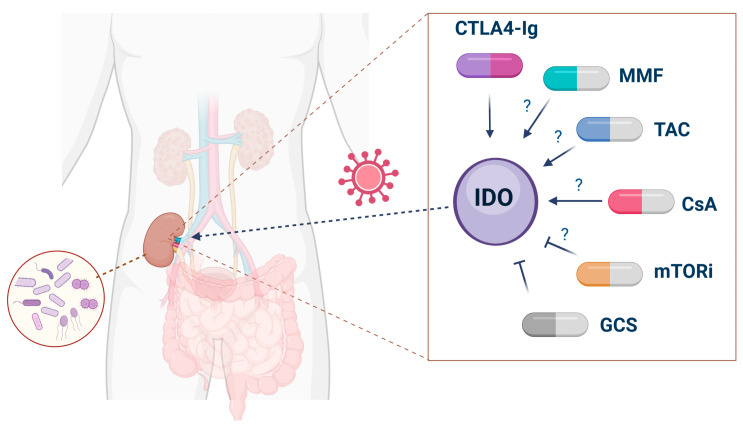
Modulation of the KYN pathway activity in kidney transplant recipients. The figure presents possible effects of immunosuppressive treatment on IDO activity, as well as other methods of KYN pathway modifications, through the impact on urinary microbiota (**left**) or local IDO transfection (**right**). CsA, cyclosporine A; CTLA4-lg, cytotoxic T-lymphocyte associated protein 4 fusion protein; GCS, glucocorticosteroids; IDO, indoleamine-2,3-dioxygenase; MMF, mycophenolate mofetil; mTORi, mammalian target of rapamycin inhibitor; TAC, tacrolimus. Created with https://www.BioRender.com (accessed on 1 August 2024).

**Table 1 ijms-25-09940-t001:** Summary of KYN pathway activity alterations and their possible effects on kidney transplant recipients. ABMR, antibody-mediated rejection; CKD, chronic kidney disease; IDO, indoleamine-2,3,-dioxygenase; KMO, kynurenine 3-monooxygenase; KYN, kynurenine; KYNA, kynurenic acid; RCC, renal cell carcinoma; Tac, tacrolimus; TCR, T-cell-mediated rejection; Trp, tryptophan.

KYN Pathway Element	Organ or System Function Modification
Indoleamine-2,3 dioxygenase (IDO)	High IDO activity correlates with chronic kidney disease (CKD) severity and inflammatory parameters [63]; IDO activity measurement as a tool for early CKD diagnosis [69]; IDO upregulation in tubular epithelial cells of rejected kidney grafts [102]; Higher IDO expression in kidney graft biopsies related to a higher incidence of antibody-mediated rejection (ABMR) or T-cell-mediated rejection (TCR) [105]; Elevated kynurenine (KYN)/tryptophan (Trp) ratio as a tool in differentiating acute graft rejection from infection [107,108]; High IDO expression in various cancers related to resistance to anticancer treatment and lower survival [133]; High IDO activity related to cardiovascular risk [23].
Kynurenine 3-monooxygenase (KMO)	Low KMO gene expression protein level [113] and activity [109] in a porcine model of kidney allograft rejection; KMO knockout results in proteinuria in animal models [110]; Low podocytic KMO expression in diabetic patients [110]; Low KMO expression in renal cell carcinoma (RCC) patients [135]; KMO inhibitors as antineoplastic drugs in certain types of cancer [142].
Kynurenic acid (KYNA)	High serum KYNA level correlates with CKD severity and inflammatory markers [63]; High plasma KYNA levels correlate with procalcitonin and lactate concentration in survivors from septic shock, whereas a lack of KYNA concentration reduction relates to poor survival [126]; Plasma KYNA concentration correlates with hyperhomocysteinemia in peritoneal dialyzed patients with atherosclerotic cardiovascular disease [159]; Serum KYNA level related to aortic stiffness in patients with atrial fibrillation [160]; Tacrolimus Tac increases KYNA production [122].

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
