# Peer review of "Kynurenine Pathway after Kidney Transplantation: Friend or Foe?"

_ijms, 2024, doi:10.3390/ijms25189940_

Round 1
Reviewer 1 Report
Comments and Suggestions for Authors
The author provides a review of the role of kynurenine pathway in patients undergoing kidney transplantation. It has significance for inflammation and immune regulation in kidney transplant recipients.
1. It is suggested that the author summarize the effects of IDO, KMO, and KYNA on transplanted kidneys or the entire body of patients (which can be presented in a table)
2. Is it possible to intervene in the abnormal activation of kynurenine pathway?
Author Response
The author provides a review of the role of kynurenine pathway in patients undergoing kidney transplantation. It has significance for inflammation and immune regulation in kidney transplant recipients.
- It is suggested that the author summarize the effects of IDO, KMO, and KYNA on transplanted kidneys or the entire body of patients (which can be presented in a table).
Thank you for your comment, Table 1 summarizing IDO, KMO and KYNA effects on organs and systems in kidney transplant recipients has been prepared as requested.
- Is it possible to intervene in the abnormal activation of kynurenine pathway?
Thank you for your comment. According to available data the inhibition of abnormal KYN pathway activity is of special importance in the treatment of malignancies or cardiovascular diseases. However generalized KYN pathway inhibition through IDO blockade increases the risk of immune mediated complications in kidney transplant recipients, especially acute graft rejection. Therefore, more selective methods of KYN pathway inhibition should be implemented. As requested, information about methods of KYN pathway inhibition has been added:
„ On the other hand, a targeted KYN pathway inhibition may provide beneficial effects on the treatment of selected complications after kidney transplantation, namely malignancies or cardiovascular diseases. Beyond KMO, KATs and KYNU inhibitors, which are potentially novel candidates in anticancer therapy [137], other already available drugs were shown to selectively inhibit KYN pathway. Angiotensin converting enzyme inhibitors [175], angiotensin II type 1 receptor blockers [176], cyclooxygenase inhibitors [177], fibrates [178] and glibenclamide [179] were reported to inhibit KATs activity, which may result in the reduction of kidney damage related complications, especially in kidney transplant recipients.“ (L489-497).
Reviewer 2 Report
Comments and Suggestions for Authors
This manuscript by Zakrocka and colleagues reviews the kynurenine pathway of tryptophan catabolism, its interactions with the host immune responses to the graft, and its impact on kidney transplant outcomes. The article also reviews the role of the pathway in short- and long-term transplant complications including carcinogenesis, infections and cardiovascular sequelae. Particular attention is focused on the inducible enzyme indoleamine 2,3-dioxygenase (IDO), which catalyzes the initial, regulated step in tryptophan catabolism, and the impact of pro- and anti-inflammatory factors on IDO activity.
This review manuscript is well-written, authoritative and up to date. It summarizes an aspect of intermediary metabolism which is impactful but underappreciated in the transplant field. This review will likely provide valuable guidance to scientists and clinicians seeking to improve kidney transplant outcomes. I have one major comment and several minor suggestions for the authors’ consideration.
Major comment: Since kidney transplant is the article’s central theme, the implications of tryptophan as a component of machine preservation solutions merits discussion. Specifically, HTK solution (where “T” represents tryptophan as a key ingredient), i.e. Custodiol, which is used by some centers for hypothermic machine preservation of deceased donor kidneys for transplant [see Lee CY & Mangino MJ, Organogenesis 2009;5:3; doi 10.4161/org.5.3.9582 and references therein], contains 2 millimolar tryptophan, which is far above physiological serum tryptophan concentrations. What is the potential impact of tryptophan metabolism in HTK-perfused kidneys? Is the tryptophan beneficial, or are there risks?
Minor comments
Lines 15-16: This statement is true only in some cases. The majority of transplant recipients, especially those receiving living donor kidneys, have good to excellent outcomes and many years of adequate graft function.
Line 24: “immune system status” is unclear. Do you mean the balance between pro- vs. anti-inflammatory immune function?
Lines 37-42: Some editing of this important introductory text is recommended: “…disorders with various etiologies and clinical presentations. Although the number of affected individuals is estimated at 850 million worldwide, the true prevalence of kidney disease may be even greater [1]. Indeed, kidney disorders rank among the most common… …people affected is double the number of diabetes patients and more than 20 times the number of human…”
Line 59: Please specify which patients – those with ESKD?
Line 67: “minimally invasive” would be more accurate than “noninvasive” since blood sampling requires sticking a needle into a peripheral vein.
Line 262: Is this the first mention of neopterin in the text? Please help the reader by explaining the significance of serum neopterin concentration.
Line 269: This appears to be the first appearance of the acronym ABMR. Please define it here.
Line 318: “…cultures were shown [114].”
Lines 352, 376: The term is “carcinogenesis.”
Line 366: If QPRT activity is elevated, but the upstream enzymes KMO and 3-HAAO are suppressed, what is the net effect on NAD+ formation in renal cell carcinoma? NAD+ formation matters in this context because it is an essential cofactor for the glycolytic enzyme GAPDH, and poorly vascularized solid tumors rely upon anaerobic glycolysis for ATP production.
Figure 3: The arrows converging on IDO imply the enzyme is induced or activated, but according to the text, at lest some of the medications suppress IDO. In that case, please indicate IDO inhibition by modifying the arrows in the same fashion as in Figure 2. Also in Figure 3 the purposes of the circle on the lower left (contains bacteria?) and the coronavirus-like symbol just to the left of IDO, are unclear. Please state in the figure legend what these elements represent, and explain how they are affected by IDO activation or suppression.
Author Response
- Major comment: Since kidney transplant is the article’s central theme, the implications of tryptophan as a component of machine preservation solutions merits discussion. Specifically, HTK solution (where “T” represents tryptophan as a key ingredient), i.e. Custodiol, which is used by some centers for hypothermic machine preservation of deceased donor kidneys for transplant [see Lee CY & Mangino MJ, Organogenesis 2009;5:3; doi 10.4161/org.5.3.9582 and references therein], contains 2 millimolar tryptophan, which is far above physiological serum tryptophan concentrations. What is the potential impact of tryptophan metabolism in HTK-perfused kidneys? Is the tryptophan beneficial, or are there risks?
Thank you very much for your comment. The impact of tryptophan available in preservation solution on graft function is a matter of debate. Lee and Mangino (Preservation methods for kidney and liver. doi: 10.4161/org.5.3.9582), as well as Zulpaite et al. (Tryptophan Metabolism via Kynurenine Pathway: Role in Solid Organ Transplantation. doi: 10.3390/ijms22041921) indicated that tryptophan may have a bidirectional role in graft survival. HTK solution has been shown comparable for kidneys when preservation times were shorter and when organs were from standard criteria donors (Bretschneider 1964, Survival time and recuperative time of the heart in normothermia and hypothermia). Tryptophan was thought to exhibit membrane protecting effect, therefore improving graft survival, and was a substrate for further metabolites, especially NAD+. Even modified HTK solution, a HTK-N solution was shown to be potentially beneficial regarding kidney graft survival, in terms of lower endothelial damage, although observations in kidney graft recipients are rare (HTK-N: Modified Histidine-Tryptophan-Ketoglutarate Solution-A Promising New Tool in Solid Organ Preservation. doi: 10.3390/ijms21186468). On the other hand, tryptophan starvation was suggested to improve graft survival, by lowering the risk of ischemia reperfusion injury through lowering the number of circulating granulocytes, decreasing the expression of genes related with oxidative stress (glutathione synthetase, catalase) or stimulating autophagy, which may decrease a secretion of proinflammatory cytokines. Since the role of tryptophan as a compound of preservation solution on kidney graft survival was not unequivocally explained, sentences regarding its possible effect on kidney graft, highlighting the need for further studies have been added:
“Importantly, Trp is present at 2 mmol/l concentration in one of the most popular preservation solutions, a histidine-tryptophan-ketoglutarate (HTK) solution, mainly because of its antioxidant and membrane stabilizing potential [14]. HTK invented by Bretschneider was shown to be a comparable preservation solution when preservation times were shorter and when standard criteria donors organs were implanted [70,71]. Modified HTK solution, a HTK-N solution, was shown to lower endothelial graft damage, although observations in kidney graft recipients are limited [72]. On the other hand, Trp deprivation is suggested to induce resistance to hypoxia and prevent ischemia-reperfusion injury, mainly by lowering the number of circulating neutrophils [73] or autophagy induction, which may decrease the secretion of proinflammatory cytokines [74]. Indeed, Eleftheriadis et al. reported that Trp deprivation increased renal proximal tubular epithelial cells survival in hypoxic conditions through autophagy induction [75]. Further studies are needed to clarify the impact of Trp on kidney graft preservation results.” (L176-188).
Minor comments
- Lines 15-16: This statement is true only in some cases. The majority of transplant recipients, especially those receiving living donor kidneys, have good to excellent outcomes and many years of adequate graft function.
Thank you very much, the sentence has been modified as requested: “However, kidney transplant recipients’ outcomes are not fully satisfactory, due to increased risk of cardiovascular diseases, infections, and malignancies.” (L15-17).
- Line 24: “immune system status” is unclear. Do you mean the balance between pro- vs. anti-inflammatory immune function?
Thank you very much, the sentence has been modified as requested: “IDO activity, as well as the production of subsequent metabolites of the pathway, is highly dependent on the balance between pro- and anti-inflammatory conditions.” (L24-25).
- Lines 37-42: Some editing of this important introductory text is recommended: “…disorders with various etiologies and clinical presentations. Although the number of affected individuals is estimated at 850 million worldwide, the true prevalence of kidney disease may be even greater [1]. Indeed, kidney disorders rank among the most common… …people affected is double the number of diabetes patients and more than 20 times the number of human…”
Thank you very much, text has been reedited as requested: “Kidney diseases represent a broad spectrum of disorders with various etiology and clinical presentation. Although the number of individuals worldwide affected by kidney diseases is estimated at 850 million, their true prevalence seems to be largely underestimated [1]. It was suggested that kidney disorders are one of the most common healthcare problems, as the number of people affected by kidney diseases is significantly larger than patients with diabetes or individuals infected by the human immunodeficiency virus (HIV) [1].” (L37-43).
- Line 59: Please specify which patients – those with ESKD?
Thank you very much, sentence has been corrected as requested: “In Europe, despite a lower transplant rate than in America, more ESKD patients live with a functioning graft (47 % vs 32 % respectively).” (L62-64).
- Line 67: “minimally invasive” would be more accurate than “noninvasive” since blood sampling requires sticking a needle into a peripheral vein.
Thank you very much for your comment. The term ‘noninvasive’ is related to graft function evaluation, and since no graft biopsy is performed, blood collection and especially urine collection are rather related with noninvasive analysis of the graft status.
- Line 262: Is this the first mention of neopterin in the text? Please help the reader by explaining the significance of serum neopterin concentration.
Thank you very much, the sentence has been corrected as requested: “Importantly, the serum KYN/Trp ratio in the rejection group was higher starting from day 1 after transplantation and correlated with serum neopterin concentration, a marker of immune system activation.” (L277-279).
- Line 269: This appears to be the first appearance of the acronym ABMR. Please define it here.
Thank you very much, but the ABMR abbreviation has been already explained in line 193.
- Line 318: “…cultures were shown [114].”
Thank you very much for the comment, the sentence has been corrected as requested: “Other effects, including inhibition of TNF-α secretion from human mononuclear cells and high mobility group box protein 1 (HMGB1) secretion in monocytes and granulo-cytes cultures were shown” (L333-335).
- Lines 352, 376: The term is “carcinogenesis.”
Thank you very much, terms have been corrected as requested. (L368, 397-398).
- Line 366: If QPRT activity is elevated, but the upstream enzymes KMO and 3-HAAO are suppressed, what is the net effect on NAD+formation in renal cell carcinoma? NAD+ formation matters in this context because it is an essential cofactor for the glycolytic enzyme GAPDH, and poorly vascularized solid tumors rely upon anaerobic glycolysis for ATP production.
Thank you very much for the comment, QPRT activity can differ in various types of cancer, being elevated (as was shown in breast cancer “A comprehensive analysis of the role of QPRT in breast cancer” doi: 10.1038/s41598-023-42566-4) or even decreased (as in kidney cancer: “Dysregulation at multiple points of the kynurenine pathway is a ubiquitous feature of renal cancer: implications for tumour immune evasion” doi: 10.1038/s41416-020-0874-y), but NAD+ can be also produced through a salvage pathway (preferred in cancer cells), from nicotinamide by nicotinamide
phosphoribosyltransferase (NAMPT). As requested, this information has been added in the text: “Other abnormalities in KYN pathway activity in tissues of RCC patients include lower expression of QPRT and downregulation of KMO and 3-HAAO [136], although QPRT activity may vary in different types of cancers [137]. Under QPRT inhibition the NAD+ synthesis is secured by the activation of a salvage pathway, often preferred by cancer cells, in which nicotinamide is converted by nicotinamide phosphoribosyltransferase (NAMPT) into NAD+ [138].” (L383-388).
- Figure 3: The arrows converging on IDO imply the enzyme is induced or activated, but according to the text, at lest some of the medications suppress IDO. In that case, please indicate IDO inhibition by modifying the arrows in the same fashion as in Figure 2. Also in Figure 3 the purposes of the circle on the lower left (contains bacteria?) and the coronavirus-like symbol just to the left of IDO, are unclear. Please state in the figure legend what these elements represent, and explain how they are affected by IDO activation or suppression.
Thank you very much, the Figure 3 and its caption have been modified as requested.